# Artificial intelligence and network science as tools to illustrate academic research evolution in interdisciplinary fields: The case of Italian design

Daniele Pretolesi[1], Ilaria Stanzani[2,3], Stefano Ravera[2,3], Andrea Vian[4], Annalisa Barla[2,3]*

**1** Center for Technology Experience, AIT - Austrian Institute of Technology, Vienna, Austria, **2** Dipartimento di Informatica, Bioingegneria, Robotica e Ingegneria dei Sistemi, Università di Genova, Genova, Italy, **3** Machine Learning Genoa Center, Università di Genova, Genova, Italy, **4** Dipartimento Architettura e Design, Università di Genova, Genova, Italy

* annalisa.barla@unige.it

**Data Availability Statement:** All data are available from https://github.com/annalisabarla/OA-ItalianDesign.

## Abstract

In this paper, we explore the application of Artificial Intelligence and network science methodologies in characterizing interdisciplinary disciplines, with a specific focus on the field of Italian design, taken as a paradigmatic example. Exploratory data analysis and the study of academic collaboration networks highlight how the field is evolving towards increased collaboration. Text analysis and semantic topic modelling identified the evolution of research interest over time, defining a ranking of pairs of keywords and three prominent research topics: User-Centric Experience Design, Innovative Product Design and Sustainable Service Design. Our results revealed a significant transformation in the field, with a shift from individual to collaborative research, as evidenced by the increasing complexity and collaboration within groups. We acknowledge the limitations faced by this work, suggesting that the methodology may be primarily suitable for bibliometric and more silos-like disciplines. However, we emphasize the urgency for the scientific community to address the future of research not indexed by large open-access databases like OpenAlex.

## Introduction

While facing an increasing number of multidisciplinary wicked problems [1, 2], science continues to operate under a traditional division into disciplines; silos that were historically structured to serve the needs of organization and specialization rather than innovation. This logic of specialization, which was once highly valuable for advancing knowledge in specific fields, now shows diminishing returns in an era defined by complexity. In a world where problems are interconnected and span multiple domains, the siloed development of knowledge limits the ability to address complex, real-world issues. Integration across disciplines is increasingly necessary, as the incremental utility of further specialization is often insufficient to tackle the challenges posed by today's multifaceted problems. Historically, integration was the

**Funding:** This work is partially funded by the European Union - NextGenerationEU and by the Ministry of University and Research (MUR), National Recovery and Resilience Plan (NRRP), Mission 4, Component 2, Investment 1.5, project "RAISE - Robotics and AI for Socio-economic Empowerment" (ECS00000035) as A. Barla is part of the RAISE Innovation Ecosystem. The funders had no role in study design, data collection and analysis, decision to publish, or preparation of the manuscript.

responsibility of decision-makers; however, artificial intelligence now offers the opportunity to apply integration at various stages of research, including problem definition, framing, modeling, and supporting problem-solving. Indeed, the definition of disciplines and their subdivisions usually trail behind the progress of science and the emergence of new discoveries. Also, they often reflect cultural influences and respond to the growing demand for specialization in various fields. In this scenario, many scientists often limit their collaborative efforts within their fields rather than seeking connections with other disciplines, limited, for instance, by the additional time and effort required to establish common ground and frameworks for interdisciplinary projects, and by how they will be evaluated during their academic careers. Nevertheless, science aims to comprehend the world we live in through a collaborative and social process [3, 4] and it is bound to do so in an ever-increasing set of complex challenges. For this reason, the pursuit of monocultural knowledge appears constrained and helpless.

Science of Science (SciSci) [5] emerges as a discipline able to provide the tools and the methods to unravel scientific complexity. Stemming from the computational domain, SciSci aims to provide to all disciplines rigorous scientific and analytical methodologies. The aim is to induce in each discipline a process of self-understanding to build their future in the academic community. Ultimately, SciSci seeks to map the evolution of science to provide a design tool of *think-thank* like initiatives to influence how our society deals with the wicked problems of our time.

In this work, we investigate how Artificial Intelligence (AI) approaches may be used to characterize interdisciplinary disciplines. In fact, this work uses data-driven approaches to, possibly, analyse any discipline, especially those where cross-fertilization is paradigmatic. This is particularly challenging when transitioning from hard sciences to social sciences and humanities which present unique challenges, particularly due to their reliance on specific scholarly output formats such as books and book chapters. Indeed, a key issue with publications and citation public databases is their emphasis on journals, often overlooking other output types of scientific knowledge dissemination like books, proceedings and reports. Additionally, these tools do not provide—yet—comprehensive geographic coverage or adequately capture the breadth and depth of subject matter in thematic journals.

As a paradigmatic case study, we decided to consider the discipline of design as it is a growing cross-boundary research field spanning from Engineering and Computer Science to Economics, Social sciences, and Arts. Here, the fragmentation across disciplines may be particularly pronounced, as reflected in the fact that even domain experts might not possess a comprehensive grasp of the entire field. This very dynamism contributes to design's rapid evolution. Furthermore, the influence of local academic and industrial contexts suggests a potentially stronger geographical influence compared to other fields. As a result, the field of design has a heightened interest in self-reflection and, consequently, a greater imperative to effectively communicate its scientific foundation. This focus on understanding and communicating its own knowledge base positions design research as a valuable case study for exploring the challenges and opportunities presented by the interdisciplinary landscape. As we shift from purely exploratory data analysis [6] towards AI-driven methods, we restrict our focus to the Italian academic design research as we can provide a direct inquiry and assessment among the Italian design community to validate the results and observed trends. Since we employ a data-driven approach, we look for repositories that possibly hold all research knowledge. Initially, we looked into IRIS [7], the official repository used by the Italian academy, which would have been the optimal source for the analysis, as all Italian faculties have to periodically submit the list of their research products into the system. Unfortunately, IRIS is not designed to collect standardized reusable data, preventing it from becoming a reliable resource for research development. Hence, we considered different platforms such as Scopus [8], WoS [9], and OpenAlex

[10]. We picked the third as it provides free APIs and metadata, and it is shaping up as a solid standard in the field of scientometrics [11]. Moreover, OpenAlex is comparable in size to Scopus and WoS in terms of collected works and authors [12].

To carry out our analysis, we leverage cutting-edge tools and methodologies from the fields of SciSci and Network Science (NetSci) to analyze scientific production data. These AI-driven approaches play a pivotal role in unravelling the Academic Collaboration Network (ACN) within the design field, facilitating a holistic examination of its multifaceted nature.

The remainder of this paper is organized as follows: first, we illustrate the state-of-the-art in the science of science for interdisciplinary research, then we describe materials, methods and the experimental setting and, finally, we illustrate the obtained results. We conclude the paper by discussing what we found and the possible implications for future works.

## Related works

Researchers have long been addressing interdisciplinarity and its role in fostering successful scientific endeavours. In [13] the assumption is that researchers are often driven towards boundary-crossing research, looking for a trade-off between high productivity versus maintaining a broad perspective. Information initiatives can offer flexibility, enabling researchers to redirect their focus from their primary specialization towards the peripheral areas that enrich their interdisciplinary work. For example, the authors of [14] define a framework of problem-solving agents, showing how diverse problem solvers can outperform groups of high-ability problem solvers. Similarly, [15] explores the relationship between interdisciplinary research and research impact, showing that higher interdisciplinarity is significantly associated with increased research impact. More recently, [16] highlights the risk of scientific *monocultures*, including the one induced by AI, which may lead to a more biased and error-prone understanding of the world hence preventing innovation.

Despite the long-standing evidence highlighting the importance of interdisciplinarity for the advancement of research, it was also found that interdisciplinary proposals tend to exhibit lower rates of funding success [17, 18] and are often perceived as high-risk proposals. Also, interdisciplinary research entails significant costs, including time investment in fostering collaborative relationships and aligning diverse perspectives. This may yield fewer and more heterogeneous research outputs compared to those from more discipline-specific efforts, at the risk of being under-evaluated by traditional evaluation metrics [19–21].

Often supported by network science [22], the study of interdisciplinary behaviours in research has been mostly conducted by focusing on two aspects: citation patterns and collaboration networks.

By representing a network of publications and citations, in [23] the authors exploit citations to explore interdisciplinary patterns in six different research fields over a span of 30 years, noting how research has mostly become a team effort and assessing how science is becoming more cross-disciplinary even if knowledge transfer appears to occur in small steps and only towards neighbouring fields.

By defining authors as nodes and collaborations as edges, network analysis unveils crucial insights into the structure and dynamics of academic collaboration [24]. Metrics like degree and betweenness centrality may help identify key players and influential groups, while community detection algorithms may reveal cohesive clusters within the network. Finally, through evolutionary analysis, researchers can track changes over time, shedding light on emerging trends and policy impacts. For example, in [25] co-authorship networks are used to predict authors' research impact.

More recently, the evolution of research trends and interdisciplinarity are studied within the current widespread use of deep learning methods and large language models in different subfields of research [26, 27], spanning from medicine [28, 29] to economy [30] to material science [31] and to artificial intelligence [32].

## Materials

To carry out our investigation, we put together a dataset of Italian design publications, starting from the official list of the 243 faculties currently affiliated with the ICAR/13 scientific disciplinary sector defined by the Italian Research Ministry (https://cercauniversita.mur.gov.it/php5/docenti/cerca.php). Each author is identified by a set of attributes including their name, last name, and affiliation. Using this list we query the OpenAlex online catalogue APIs, retrieving the list of all their publications. Evoking the renowned Library of Alexandria, a cultural and literary beacon of the ancient world, OpenAlex emerged as a successor to Microsoft Academic Graph (MAG) [33], a vast repository of scientific research publications that ceased operations on December 31, 2021. Recognizing the significance of unfettered access to knowledge, the architects of OpenAlex opted to adopt and adapt MAG's models, creating a freely available and universally accessible database unshackled from commercial constraints. OpenAlex's raison d'être, in essence, is the dissemination of knowledge. With over 240 million works readily accessible, OpenAlex expands daily with approximately 50,000 new data points. This immense body of knowledge is meticulously organized into a heterogeneous and directed graph, employing eight distinct node types:

1. Works: Encompassing abstracts of articles, books, patents, datasets, and theses, these entities represent the foundation of scholarly output.

2. Authors: Every individual contributing to the creation of a work. Note that each author is complemented with an affiliation field listing all known affiliations throughout the years. The affiliations have a country field which allows us to associate the publication with a country of interest.

3. Sources: Journals, archives, and other repositories preserving works form the backbone of this category.

4. Institutions: Universities, research centers, and organizations where authors hold affiliations.

5. Concepts: Abstract ideas addressed in various articles are aptly categorized under hierarchical concepts, with OpenAlex assigning approximately 65,000 of these labels to each work.

6. Publishers: Companies and organizations responsible for disseminating works.

7. Research Funders: Those who provide financial support for research endeavours.

8. Geographic Areas: The locations where authors conduct their research or where works are produced.

## Methods

### Data handling

Our data collection process began with identifying relevant researchers using the OpenAlex search by name API. To account for potential name ambiguity, we employed strategies that

**Table 1. Design concepts identified by experts.**

**Concepts related to design**

*Art and design, Book design, Co-design, Collaborative design, Communication design, Conceptual design, Critical design, Design and Technology, Design brief, Design cycle, Design education, Design elements and principles, Design for All, Design for the Environment, Design for X, Design history, Design knowledge, Design language, Design methods, Design strategy, Design technology, Design thinking, Design tool, Design-based research, Design for manufacturability, Design management, Design process, Design science, Design science research, Ecological design, Environmental design, Environmental design and planning, Environmental graphic design, Evidence-based design, Experience design, Fashion design, Generative Design, Graphic design, Human Computer Interaction, Industrial design, Information design, Instructional design, Interaction design, Interactive design, Interface design, Interior design, Iterative design, Integrated design, Material Design, Package design, Parametric design, Participatory design, Philosophy of design, Product design, Product design specification, Rollover (web design), Service design, Spatial design, Strategic design, Sustainable design, Textile design, Universal design, User centered design, User experience design, User interface design, User-centered design, Web design, Website design*

**Table 2. Publication type split.**

| Publication Type | Number of publications |
| --- | --- |
| article | 604 |
| book-chapter | 210 |
| book | 17 |
| editorial | 1 |
| other | 1 |
| peer-review | 1 |

considered researchers' affiliations. This initial search yielded a total of 5317 publications. Following duplicate removal, the corpus was reduced to 4503 works. To refine our focus, we further restricted the publication window to years with at least 10 published works, resulting in a dataset spanning from 2000 to 2024 and containing 4365 publications. Of these, we kept only the publications that are associated to at least one concept among a list of concepts characterizing the field of design. The list was obtained by a pool of academic researchers in the field, including one of our co-authors, who selected a set of 68 relevant concepts within the OpenAlex catalogue, listed in Table 1. We then extended the list following the OpenAlex concept hierarchy, incorporating all children nodes of the initial set, leading to a total of 230 concepts. After this filter, we were left with 860 publications.

After, we removed all publications that were not written in English and where the publication type was below a 1% threshold and obtained a total of 834 works. In Table 2 we provide an overview of the types and frequency of works included in the dataset.

For the text analysis, we further excluded publications missing titles or abstracts, leading to a dataset of 708 works. Lastly, before proceeding with the analysis, we explored the lengths of abstracts. The gathered statistics provided crucial insights into the distribution of abstract lengths. Notably, we identified that the 5th and 95th percentiles of abstract lengths were 54 and 372 words, respectively. To prevent biased results, we excluded works associated with abstract lengths that fell beyond the two thresholds, whether shorter or longer, narrowing the dataset to a total of 637 works were taken into account.

## Heterogeneous graphs as a representation of academic collaboration networks

Heterogeneous Graphs [34], are very efficient abstractions for complex datasets that model data as a graph allowing the presence of multiple types of nodes and/or edges. As depicted in

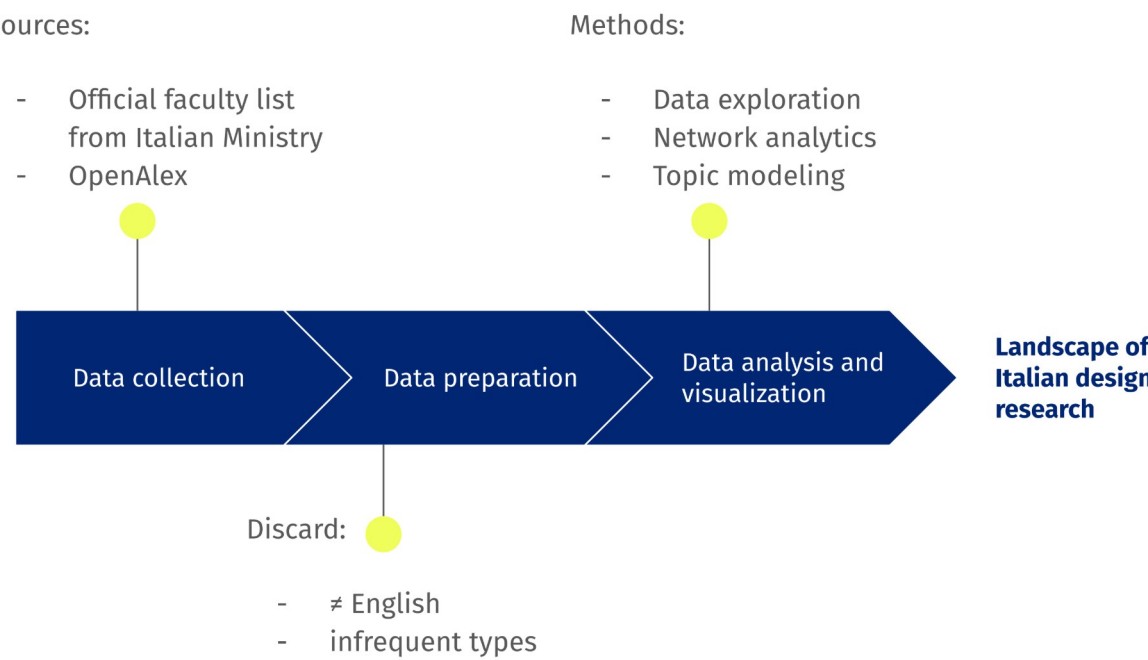

**Fig 1. Scheme of the heterogeneous graph, connecting authors (A), publications (P), years (Y), and concepts (C).**

Fig 1, an ACN may be represented as a heterogeneous graph with authors (A), papers (P), years (Y) and concepts (C) as nodes, wherein edges indicate the "co-authorship" (A–A), "write" (A–P), "discusses of" (P-C), "published in" (P-Y) relationships. The characterisation and modelling of ACNs is an ongoing and open problem relevant to understanding how scientific research is evolving and adapting to the ever-increasing complexity of our world [35]. Once we built the heterogeneous graph G, we selected the homogeneous sub-graph of authors, where we only consider links representing co-authorships. To take into account the temporal evolution, we also considered the temporal variable, by generating a set of heterogeneous sub-graphs, once every 5 years, as shown in Fig 7. Similarly, we also considered the corresponding co-authorship sub-graphs to elucidate how the collaboration among design researchers has evolved.

To evaluate such complex structures we resorted to several state-of-the-art metrics [22]. The density of a graph ranges from 0 to 1. A density of 0 means there are no edges in the graph, while a density of 1 means that every possible edge is present in the graph. Density provides insight into how many connections exist in a graph relative to the total number of possible connections. Higher density indicates a more densely connected graph, while lower density suggests a sparser one.

The average degree provides a measure of the overall connectivity of the graph. A higher average degree indicates that, on average, each vertex has more connections, while a lower average degree indicates fewer connections per vertex.

In graph theory, transitivity is a measure of how interconnected a graph is, specifically in terms of the existence of triangles within the graph. The transitivity of a graph ranges from 0 to 1. A transitivity of 1 means that every connected triple in the graph forms a triangle, while a transitivity of 0 means that there are no triangles in the graph.

The clustering coefficient for an individual node measures the likelihood that its neighbours are also connected to each other. The average clustering coefficient of a graph is then the average of these individual clustering coefficients across all nodes in the graph.

## Text analysis and semantic topic modeling

Moving from the exploratory analysis of the dataset towards a semantic approach, we narrowed our focus to publications with accessible abstracts. This approach aims to explore the evolution of the research interests of the entire field over time. In doing so, we will provide objective results of the changing trends that have characterised the last 25 years of Italian design research. We proceeded with two experiments, one based on standard Natural Language Processing (NLP) methods, and the other exploiting deep learning methods and large language models (LLM).

**Bigram ranking over time.** In NLP, an N-gram refers to a consecutive sequence of n items (or units) extracted from a particular sample of text or speech. N-grams hold significant importance in text mining and diverse applications within NLP. They encapsulate a set of words that commonly co-occur within a defined context. N-grams find extensive application in computational linguistics, serving various purposes such as text analysis, language modelling, and machine learning [36]. Here, we defined the top bigrams for the subset of publications within a given timespan of 5 years, ranked according to their frequency. Using data visualization, we built a bump chart plot illustrating how bigrams have evolved over time and possibly shading light into the evolution of design research.

**Semantic topic modeling.** To capture the essence of the academic research, we proceeded by setting up a semantic topic modelling problem that takes as input an embedding that is a combination of Bidirectional Encoder Representations for Transformers Model (BERT) [37] and Latent Dirichlet Analysis (LDA) [38, 39].

BERT is a transformer-based deep learning model pretrained by Google. It provides a robust representation of the semantic content of documents. For each publication, the title and abstract were fed into the pre-trained BERT model to derive fixed-size embeddings. At the same time, we used LDA to infer a more contextual embedding that may capture the nuances of individual themes within each publication. The concatenated embedding vectors, weighted according to a hyperparameter to balance their relative significance, were then fed into an autoencoder to reduce input dimensionality forcing the encoder to compress the input data into a smaller latent space [40]. This let us obtain a latent representation that capture the most important features or patterns in the data, while discarding the noise or redundancy.

Finally, we set up a clustering problem in the latent space, which we solved with KMeans [41]. The optimal number of clusters *K* was determined among a set of values based on the maximization of the *silhouette* score, a metric that assesses the quality of the clustering by measuring the cohesion within clusters and separation between clusters [42]. A higher silhouette score indicates that the object is well-matched to its own cluster and poorly-matched to neighbouring clusters, suggesting a good clustering configuration.

This resulted in a list of topics, each represented by a set of keywords. To provide meaningful labels for these topics, we identified the top 20 most frequent words within each topic and leveraged chatGPT, the chatbot service powered by the GPT language model from OpenAI [43], to generate appropriate names based on such keywords. Finally, we measured researchers' interest in these topics by counting how many publications were assigned to each topic across all years within the considered time span.

The pipeline (See Fig 2) is inspired by the work of [44]. It was implemented in Python exploiting the L4 GPU High-RAM hardware on Google Colab. The data used in this project are available on GitHub (https://github.com/annalisabarla/OA-ItalianDesign).

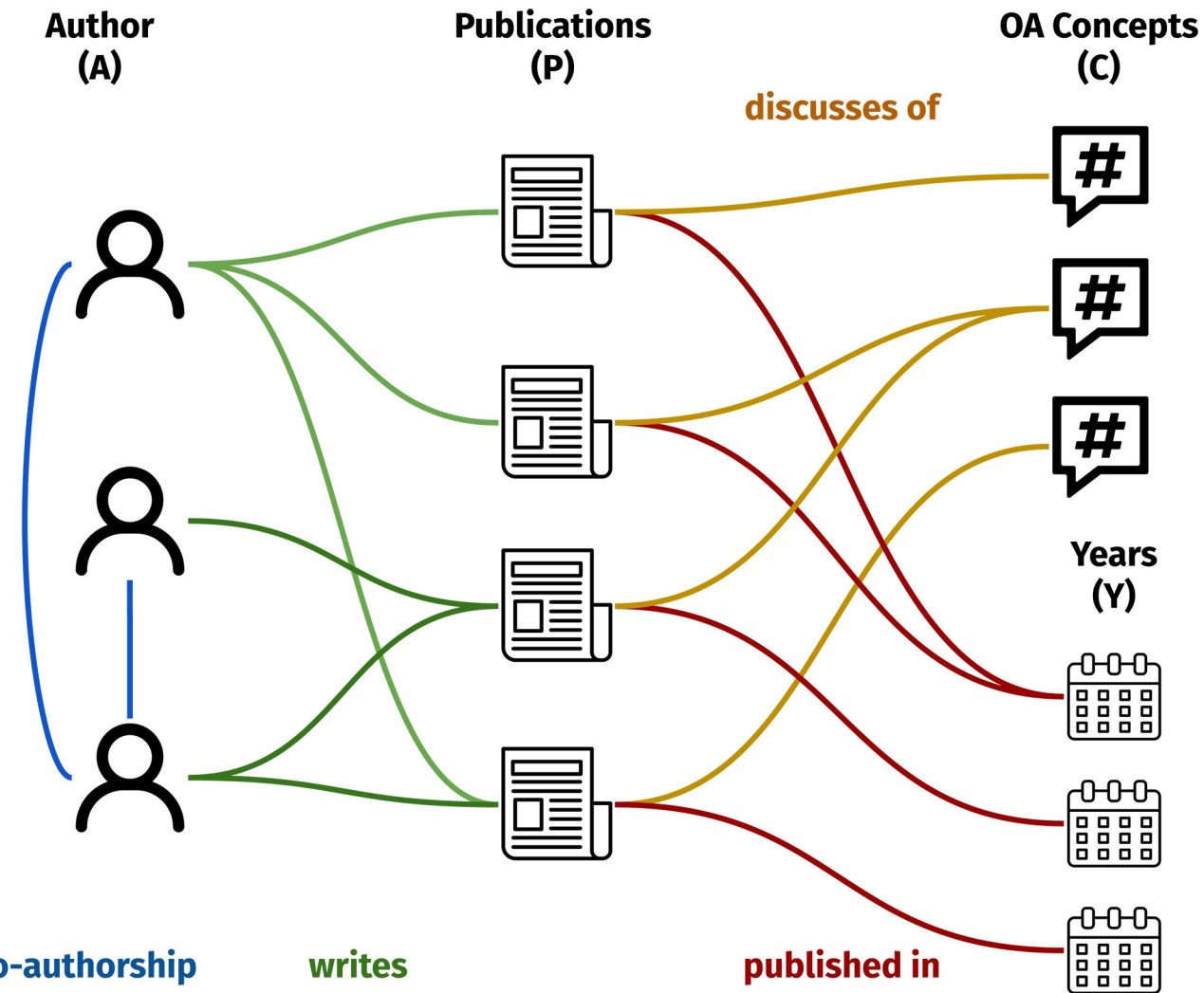

**Fig 2. Overview of the pipeline used in this work, highlighting sources and methods to characterise the landscape of Italian design research.**

## Results

### Data characterisation

Through exploratory data analysis and visual representation, we examined the dataset to gain a comprehensive insight into the collection of design publications and the possible connections among variables. Fig 3 illustrates the yearly distribution of publications, with a visible surging trend since 2000.

Fig 4 depicts the annual publication count, showcasing articles, books and book chapters as the categories with significant representation, defined as those contributing over 1% to the overall publication corpus. The cut-off date in both these figures is set to 2025, since few journal articles in our dataset are already in a preprint stage. Consequently, our analysis focused exclusively on these publication types. Fig 5 illustrates a heatmap delineating the progression of the median author count across different publication types. For articles and book chapters, we noted a steady increase in the number of authors as their series are associated with a positive linear regression coefficient (0.04 and 0.03, respectively).

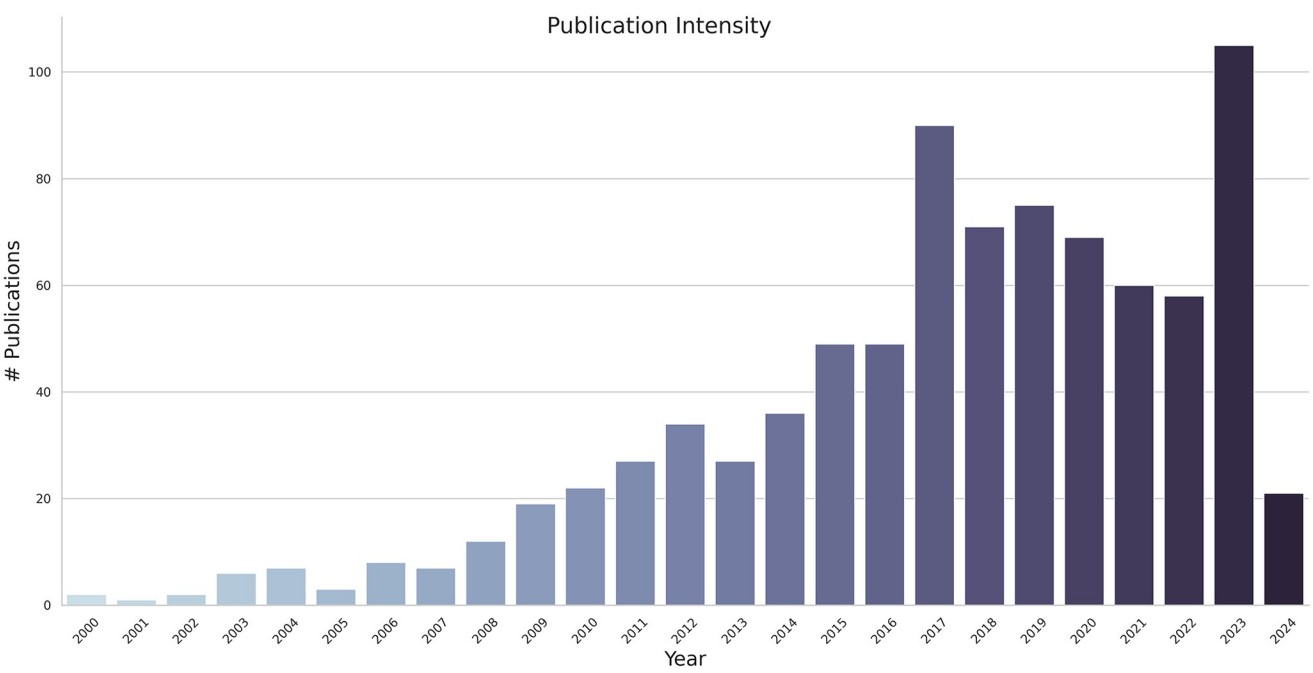

**Fig 3. Publication intensity over time.**

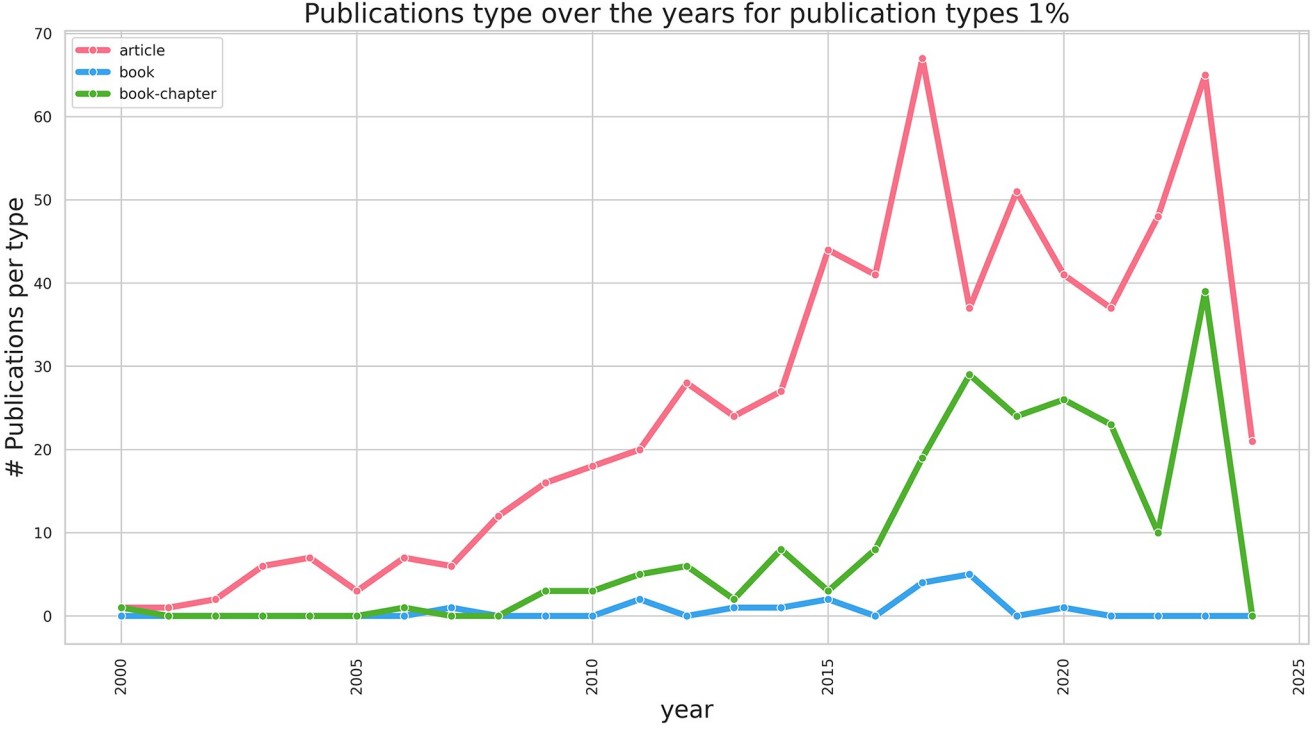

**Fig 4. Publications type trend above 1%.**

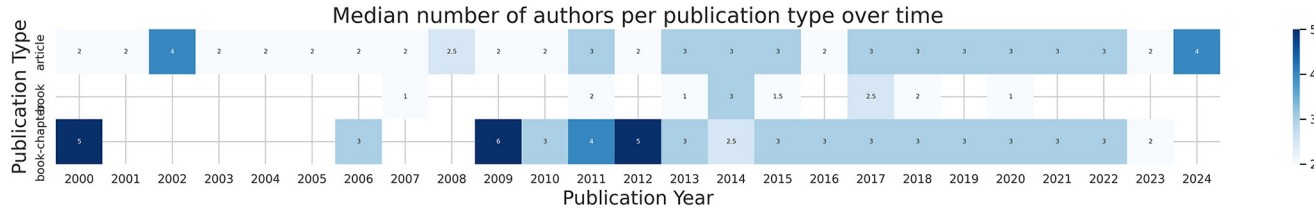

**Fig 5. Publication type over time.**

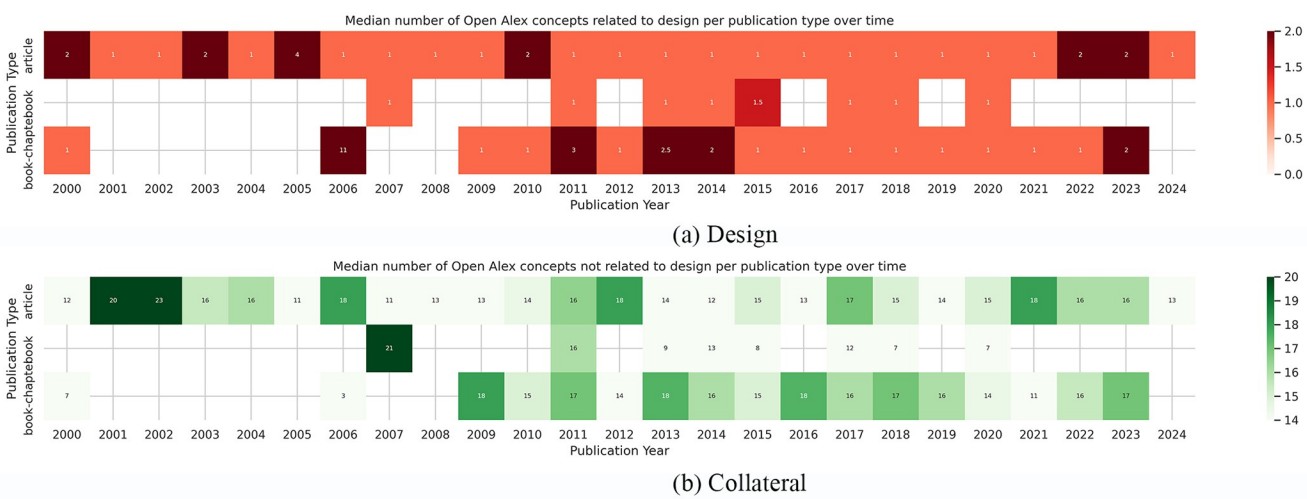

**Fig 6. Average number of OpenAlex concepts per publication type over time: (a) concepts related to design, and (b) collateral concepts.**

We then explored the relationship between the concepts that OpenAlex assigns to each publication and the design domain. We present the results in Fig 6. The top panel shows a heatmap displaying the average number of design-related OpenAlex concepts per publication type over time, whereas the bottom panel illustrates a heatmap considering collateral interdisciplinary concepts. To define concepts related to design, we used the same criterion used to filter the publications in the preprocessing, referring to the concepts listed in Table 1.

## Design landscape with network analysis and graph structures

After data exploration, we exploited graph structure to devise how collaborations have evolved over time in the landscape of Italian design. To this aim, we considered a set of 5-year long periods and, for each, we built a heterogeneous graph, following the schema in Fig 1. The heterogeneous graphs are shown in Fig 7, while the statistics computed on each graph are reported in Table 3.

For each ACN we then extracted the co-authorship subgraph, which considers only the authors and the author-author edges and consists of several connected components. This allows us to better understand the evolution of collaboration patterns. We report the corresponding network analytics statistics in Table 4. Finally, we computed the distribution of the number of authors per connected component over time, as shown in Fig 8. This highlights a pattern of increasingly bigger groups: the median value of co-authors per CC increases as well as the maximum observed value.

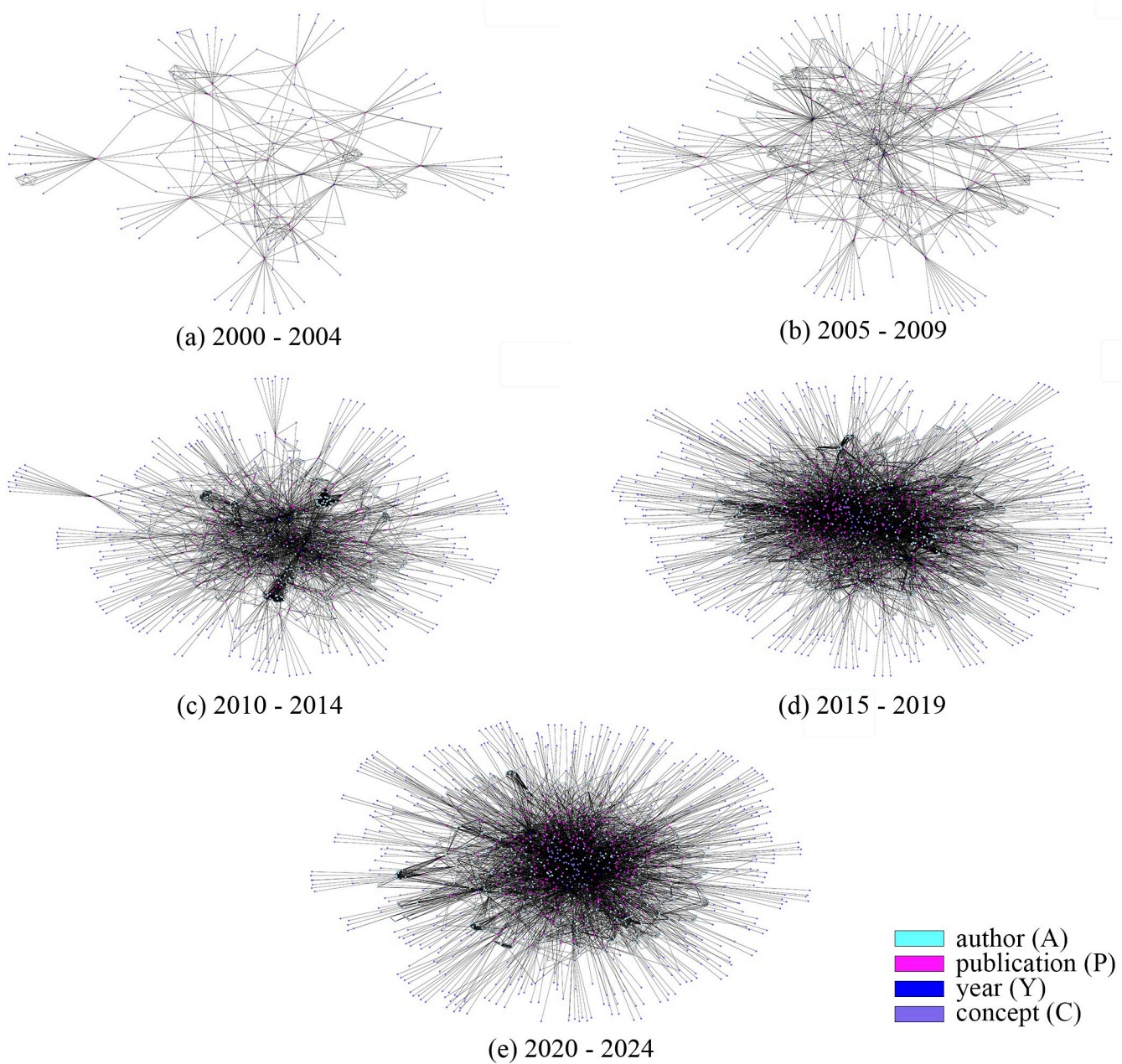

(a) 2000 - 2004

(b) 2005 - 2009

(c) 2010 - 2014

(d) 2015 - 2019

(e) 2020 - 2024

author (A)
publication (P)
year (Y)
concept (C)

**Fig 7. Heterogeneous graphs of ACNs in Italian design over the past 25 years.** Each graph displays the network of publications, authors, years, and concepts.

## Painting the landscape of research interests evolution over time

The next part of our analysis consisted of exploiting state-of-the-art NLP to devise the evolution of research interests among the design community.

First, we considered the subset of publications for which the abstract is available. For the same 5 time intervals defined above, we proceeded with extracting the most relevant bigrams, which are the most frequently occurring word pairs independently of their order. We illustrate how the top-20 bigrams change over time in a *bumpchart* plot in Fig 9.

**Table 3. Summary of heterogeneous graph analytics for different time periods.**

| Heterogeneous Graph Analytics | | | | | |
|---|---|---|---|---|---|
| | **2000-2004** | **2005-2009** | **2010-2014** | **2015-2019** | **2020-2024** |
| **Nodes** | 162 | 341 | 886 | 1552 | 1521 |
| **Edges** | 344 | 836 | 4113 | 7434 | 6742 |
| **Components** | 1 | 1 | 1 | 1 | 1 |
| **Density** | 0.0264 | 0.0144 | 0.0105 | 0.0062 | 0.0058 |
| **Avg. Degree** | 4.246 | 4.9032 | 9.2844 | 9.5799 | 8.8652 |
| **Transitivity** | 0.1997 | 0.1616 | 0.3823 | 0.1291 | 0.1026 |
| **Avg. Clust. Coeff.** | 0.2607 | 0.2414 | 0.3292 | 0.3457 | 0.3308 |

**Table 4. Summary of graph analytics for different time periods.**

| Authors Graph Analytics | | | | | |
|---|---|---|---|---|---|
| | **2000-2004** | **2005-2009** | **2010-2014** | **2015-2019** | **2020-2024** |
| **Nodes** | 40 | 88 | 316 | 592 | 562 |
| **Edges** | 52 | 138 | 1537 | 2027 | 1667 |
| **Components** | 13 | 21 | 47 | 66 | 71 |
| **Density** | 0.0667 | 0.0361 | 0.0116 | 0.0125 | 0.0106 |
| **Avg. Degree** | 2.6 | 3.1364 | 9.7278 | 6.848 | 5.9324 |
| **Transitivity** | 0.984 | 0.7516 | 0.9189 | 0.8273 | 0.8207 |
| **Avg. Clust. Coeff.** | 0.6833 | 0.6691 | 0.8055 | 0.78 | 0.7848 |

Then, we set up a topic modelling experiment where the goal was to consider all publications in the subset and assign each of them to a cluster representing a topic. We take as input title and abstract textual information, proceed with embedding them with two complementary approaches, and find a latent representation in a smaller space, where we solve the clustering problem. The process identified three topics: User-Centric Experience Design, Innovative Product Design and Sustainable Service Design. We visualized the prevalence of publications per topic over time in Fig 10.

**Fig 8. Distributions of the number of authors per connected component over time.**

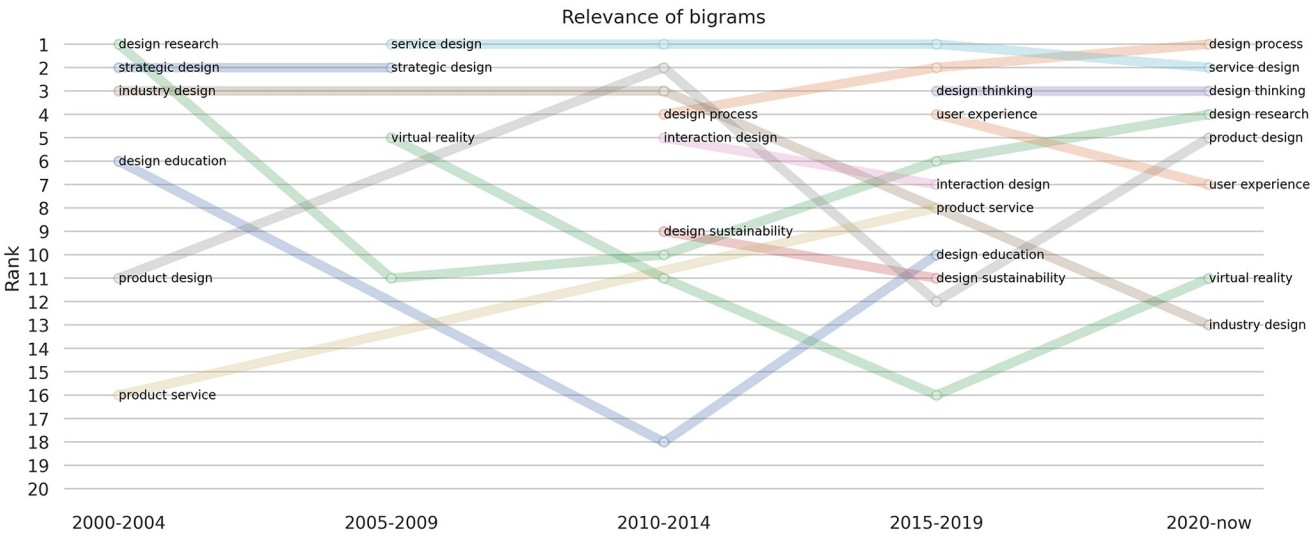

**Fig 9. Bumpchart plot displaying the bigram ranking variation over 5 decades from 2000 to 2024.**

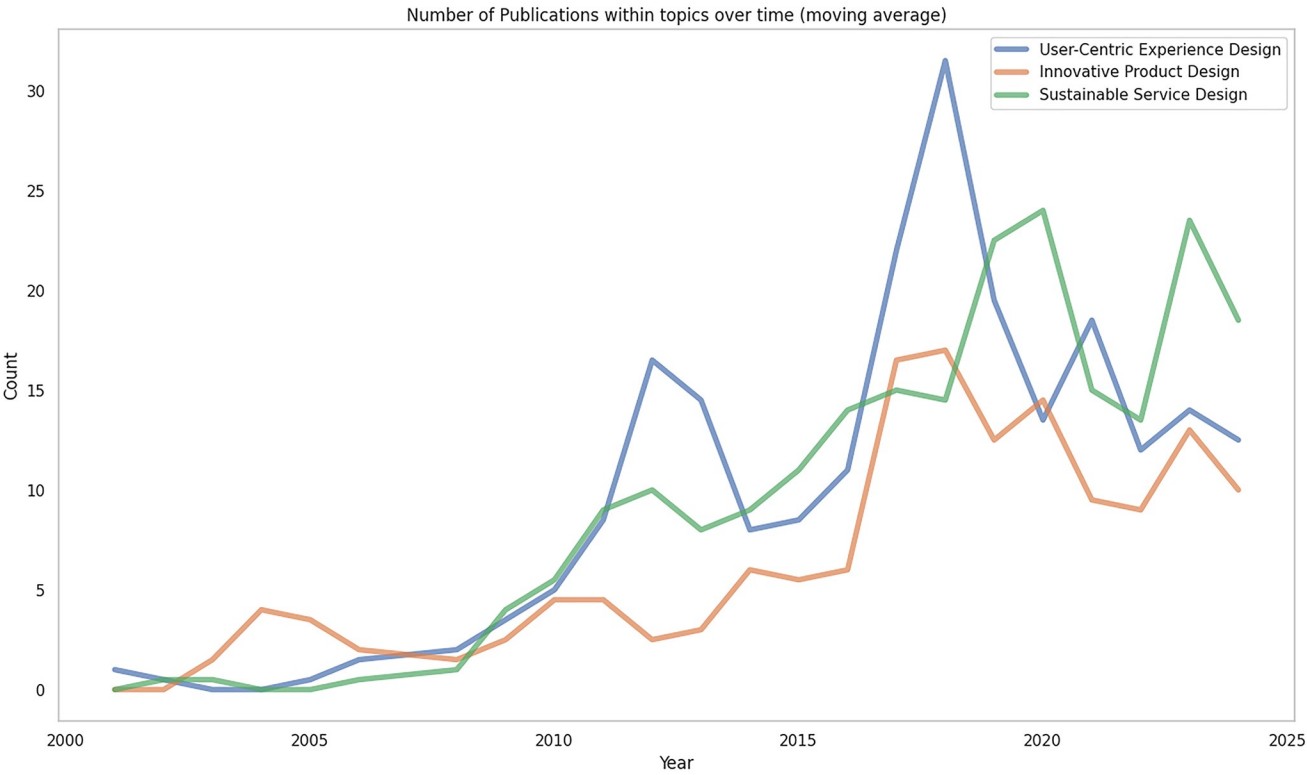

**Fig 10. Line chart plot displaying the number of publications per topic over 5 decades from 2000 to 2024.**

## Discussion

Our study provides evidence that the field of design is undergoing a significant transformation. The increasing complexity and collaboration within groups suggest a shift from individual researchers conducting studies to larger groups working together. This is further supported by

the positive linear regression coefficient (Fig 5 and Table 4), which may indicate a transition towards collaborative research.

Contrary to our initial expectations, we did not observe an increase in the number of concepts per publication over time. This could be attributed to the limited timeframe of our study, induced, in turn by the scarce availability of publications. Another potential explanation could be the limitations of OpenAlex in accurately assigning concepts, possibly due to a lack of comprehensive data on this subject.

The analysis of the heterogeneous graph structure of the Author Collaboration Network (ACN) and the distribution of authors per connected component (Figs 7 and 8) clearly shows an increase in groups of authors collaborating together. These findings are further corroborated by the graph analytics reported in Table 4.

Our topic modelling analysis identified three prominent clusters of related concepts: User-Centric Experience Design, Innovative Product Design and Sustainable Service Design. These clusters also align with the findings of the bigram ranking variation (Fig 9). For instance, the number of publications over time (see Fig 10) in User-Centric Experience Design between 2015 and 2020 is mirrored by the appearance of keywords such as *'user experience'* and *'design thinking'* on the plot. Similarly, the descending trend in Sustainable Service Design is reflected by the slight decrease of the *'design sustainability'* keyword. Lastly, changes in the Innovative Product Design cluster are almost exactly mirrored by changes in keywords such as *'product design'*, *'product service'*, and *'virtual reality'*.

In conclusion, our findings suggest a paradigm shift in the field of design towards more collaborative and group-oriented research. However, further studies are needed to confirm these trends and explore their implications in greater depth.

## Limitations

This scientific work acknowledges several limitations. Certain disciplines, particularly in the humanities, heavily favour specific scholarly work formats such as books and book chapters. Regrettably, these formats are not adequately tracked by tools like OpenAlex, WoS, and Scopus. Furthermore, the coverage of humanities and social science journals within these tools lacks comprehensiveness in terms of subject variety and depth.

OpenAlex, despite its high coverage of publications over time and comparable coverage with Scopus and WoS in the past decade, has its limitations. The concepts provided by OpenAlex curators are limited and not accurate, originating from MAG. These are being replaced by more accurate and complex metadata known as Topics and Fields. The coverage of affiliation fields is poor, and the identification process for authors is still ongoing.

The dataset, originating from the Ministry of University and Research (MUR) faculty list, incorporates some bias. It inevitably favours the present and recent past over the distant past, considering only those academics currently in the workforce for Italian universities. The identifiers used are limited to name, last name, and current affiliation, as MUR does not share more reliable identifiers such as ORCID [45]. This results in many namesakes and necessitates filtering using a set of arbitrarily chosen concepts related to design. Consequently, the dataset is very narrow and much smaller than initially expected. This concludes the limitations of this scientific work.

## Conclusions

This study presents the promises and limits of data-driven approaches in describing the evolution of interdisciplinary research, with a comprehensive view of the scientific collaboration landscape within Italian design research. It specifically highlights the growth of design into

broader communities of researchers, focusing on a unique range of topics that have evolved over time.

We have shown how a pipeline that combines data-driven and network science approaches can indeed provide a cohesive mapping of research. Looking ahead, we can imagine that with the help of Deep Learning and improvements in the OpenAlex data structure, this task could yield even more detailed representations of an interdisciplinary research field. Moreover, AI presents a dual opportunity: it can both strengthen the integration of knowledge across disciplines by uncovering new connections and patterns, while also enhancing specialization by enabling more refined and focused analysis within specific domains. This flexibility positions AI as a powerful tool in navigating the complexity of modern research.

While this work examines the discipline of design as a whole, the long-term goal is to track the evolution of trends to better understand interdisciplinary scientific production and impact, even for individual authors. In trying to balance the effects of an increase in scientific production (i.e., quantity over quality), objective metrics could also be identified in research contexts where citations are not used to evaluate scientific production. Therefore, the scientific community should strive to be able to analyze and understand the evolution of the disciplines that make it up.

Although our investigation effectively depicted this scenario, we acknowledge that the limitations we faced may make this methodology primarily suitable for bibliometric disciplines, such as computer science or medicine. However, we believe it is crucial for the scientific community to urgently address the question of the future of research not indexed by large open-access databases like OpenAlex.

## Author Contributions

**Conceptualization:** Daniele Pretolesi, Andrea Vian, Annalisa Barla.

**Data curation:** Ilaria Stanzani, Stefano Ravera.

**Methodology:** Daniele Pretolesi, Andrea Vian, Annalisa Barla.

**Software:** Stefano Ravera, Annalisa Barla.

**Supervision:** Andrea Vian, Annalisa Barla.

**Writing – original draft:** Daniele Pretolesi, Annalisa Barla.

**Writing – review & editing:** Daniele Pretolesi, Ilaria Stanzani, Annalisa Barla.

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
