## [Decision Letter · Decision Letter 0]

1 Sep 2024

PONE-D-24-17407Artificial intelligence and network science as tools to illustrate academic research evolution in interdisciplinary fields: the case of Italian designPLOS ONE

Dear Dr. Barla,

Thank you for submitting your manuscript to PLOS ONE. After careful consideration, we feel that it has merit but does not fully meet PLOS ONE’s publication criteria as it currently stands. Therefore, we invite you to submit a revised version of the manuscript that addresses the points raised during the review process.

We look forward to receiving your revised manuscript.

Kind regards,

Diego R. Amancio

Academic Editor

PLOS ONE

“This work is partially funded by the European Union - NextGenerationEU and by the Ministry of University and Research (MUR), National Recovery and Resilience Plan (NRRP), Mission 4, Component 2, Investment 1.5, project “RAISE - Robotics and AI for Socio-economic Empowerment” (ECS00000035) as A. Barla is part of the RAISE Innovation Ecosystem.”

“This work is partially funded by the European Union - NextGenerationEU and by the Ministry of University and Research (MUR), National Recovery and Resilience Plan (NRRP), Mission 4, Component 2, Investment 1.5, project “RAISE - Robotics and AI for Socio-economic Empowerment” (ECS00000035) as A. Barla is part of the RAISE Innovation Ecosystem.”

“This work is partially funded by the European Union - NextGenerationEU and by the Ministry of University and Research (MUR), National Recovery and Resilience Plan (NRRP), Mission 4, Component 2, Investment 1.5, project “RAISE - Robotics and AI for Socio-economic Empowerment” (ECS00000035) as A. Barla is part of the RAISE Innovation Ecosystem.”

Reviewers' comments:

Reviewer's Responses to Questions

**Comments to the Author**

1. Is the manuscript technically sound, and do the data support the conclusions?

Reviewer #1: Yes

Reviewer #2: Yes

2. Has the statistical analysis been performed appropriately and rigorously? 

Reviewer #1: N/A

Reviewer #2: N/A

3. Have the authors made all data underlying the findings in their manuscript fully available?

Reviewer #1: Yes

Reviewer #2: Yes

4. Is the manuscript presented in an intelligible fashion and written in standard English?

Reviewer #1: Yes

Reviewer #2: Yes

5. Review Comments to the Author

Reviewer #1: The article is very interesting and applies a new way of analysing the literature on a subject. In general, I believe it fulfils the scope of the Journal. However, I would like to make a few specific observations:

1) Firstly, I missed a figure summarising for the reader the method used, I believe this would have greatly enriched the article; 2) in the description of the research sources, it was not clear which sources were chosen and which were not. I believe that in order to reproduce the study, it is essential to identify the sources researched, such as WoS, Scopus, etc. The method needs to be described so that other researchers interested in the topic can reproduce it with the same parameters indicated by the authors. 3) In Figures 2 and 3, include the cut-off date for the last year of research. This could be given in a note or in the figure itself. 4) Figure 6 is difficult for the reader to understand what information you want to express. 5) In general, the quality of the figures needs to be improved.

I wish you a good review

Reviewer

Reviewer #2: Dear Authors,

I really appreciate your paper. To improve it please find my comments below:

1. IT would be helpful to explain the logic underlining the "silos development of knowledge". I mean some lines about the logic of specialization that prevailed over the logic of integration. nowadays, the incremental utility of specialization is low in a society characterized by complexity.

2. I suggest to clarify that in the past the integration was a competence/responsibility of decision makers. Artificial intelligence create the opportunity to apply integration at different steps: problem definition, framing, modelling, support to problem solving.

3. I also suggest to clarify that interdisciplinary approach can be used both for design of algorithms and data collection to train artificial intelligence.

4. If i did not lost it, i suggest to say something about the characteristic of different kind of artificial intelligence, I mean non supervised, supervised, machine learning and deep learning. What are the implications for your case?

5. Do you think that artificial intelligence is helpful to strengthen the integration dimension or also to strengthen specialization/vertical dimension of knowledge? Please clarify these aspects.

6. I suggest to say something about the potential misuse of artificial intelligence in your case. I mean potential bias or opportunistic use. How is it possible to prevent this potential misuse?

6. PLOS authors have the option to publish the peer review history of their article (what does this mean?). If published, this will include your full peer review and any attached files.

Reviewer #1: **Yes: **Dr. Marcio Pereira Basilio

Reviewer #2: **Yes: **Elio Borgonovi

---

## [Author Response · Author response to Decision Letter 0]

14 Oct 2024

Dear Reviewers,

Thank you for your thoughtful feedback on our article. We appreciate the opportunity to address your concerns and provide additional clarity on certain aspects of our research.

Reviewer 1

R1: Firstly, I missed a figure summarising for the reader the method used, I believe this would have greatly enriched the article;

A figure showing our methodology has been added to the work.

R1: In the description of the research sources, it was not clear which sources were chosen and which were not. I believe that in order to reproduce the study, it is essential to identify the sources researched, such as WoS, Scopus, etc. The method needs to be described so that other researchers interested in the topic can reproduce it with the same parameters indicated by the authors. 

We appreciate your emphasis on the importance of transparency in describing research sources to ensure reproducibility. In our manuscript, we explicitly stated that OpenAlex was the primary research source, and the rationale behind its selection is explained in detail at the end of the introduction. Additionally, in the materials section, we provided further information on how we leveraged the OpenAlex repository, including the specific parameters used for data extraction. We believe these sections collectively provide a clear explanation of both the chosen research sources and the methodology used to ensure that other researchers can reproduce the study.

R1: In Figures 2 and 3, include the cut-off date for the last year of research. This could be given in a note or in the figure itself. 

When Figures 2 and 3 are introduced, a sentence was added providing some context about the cut-off date of our visualisations.

R1: Figure 6 is difficult for the reader to understand what information you want to express. 

Figure 6 has now been improved by bringing forward the legend which was previously too small to read.

Reviewer 2

R2: IT would be helpful to explain the logic underlining the "silos development of knowledge". I mean some lines about the logic of specialization that prevailed over the logic of integration. nowadays, the incremental utility of specialization is low in a society characterized by complexity.

We agree that it is important to provide a clearer explanation of the logic underpinning the "silos development of knowledge." In response, we have revised the Introduction section of the manuscript to better explain how the traditional division into disciplines reflects a historical need for specialisation and organisation. However, in the context of contemporary complex, multidisciplinary challenges, this approach is becoming less effective. We have emphasised that while specialisation was crucial for past scientific advancement, it now offers diminishing utility in addressing today's interconnected and complex problems. This revision highlights the importance of interdisciplinary collaboration and integration in modern scientific inquiry.

R2: I suggest to clarify that in the past the integration was a competence/responsibility of decision makers. Artificial intelligence create the opportunity to apply integration at different steps: problem definition, framing, modelling, support to problem solving.

In response to your feedback, we have revised the introduction to include this clarification. We now highlight that AI allows integration to occur at various stages such as problem definition, framing, modelling, and supporting problem-solving. This addition emphasises the evolving role of AI in addressing multidisciplinary challenges and improving integration throughout the research process.

R2: If i did not lost it, i suggest to say something about the characteristic of different kind of artificial intelligence, I mean non supervised, supervised, machine learning and deep learning. What are the implications for your case?

While we recognize the importance of these AI distinctions, we believe that a detailed exploration of these topics would fall outside the scope of this paper, which is focused on network science. The core of our study is not directly related to the various AI methodologies but rather the application of network science principles. Therefore, we feel that including this additional discussion would not significantly enhance the quality or focus of the work.

R2: Do you think that artificial intelligence is helpful to strengthen the integration dimension or also to strengthen specialization/vertical dimension of knowledge? Please clarify these aspects.

We have revised the conclusion to include this perspective. Specifically, we have added a sentence highlighting that AI can enhance integration by uncovering new connections and patterns across disciplines while also supporting specialisation by enabling more detailed and focused analyses within specific domains. This dual capability of AI provides valuable tools for navigating and addressing the complexities of modern research.

R2: I suggest to say something about the potential misuse of artificial intelligence in your case. I mean potential bias or opportunistic use. How is it possible to prevent this potential misuse?

We understand the importance of addressing potential risks associated with the use of artificial intelligence (AI), such as bias or opportunistic use. However, in the context of our study, which employs AI and network science as tools to illustrate the evolution of academic research in interdisciplinary fields, we do not foresee significant risks of misuse. Our focus is on leveraging these tools to provide a clearer understanding of research trends and collaboration patterns.

That said, we acknowledge that AI can present challenges related to bias and misuse in broader applications. While these concerns are important, they fall outside the scope of our current research, which primarily aims to demonstrate methodological applications rather than addressing these broader ethical considerations. We appreciate your understanding and hope this clarification aligns with the focus of our work.

Once again, we appreciate your thoughtful reviews and feedback, and we are committed to addressing any further concerns or suggestions you may have.

Sincerely,

The Authors

---

## [Decision Letter · Decision Letter 1]

22 Nov 2024

Artificial intelligence and network science as tools to illustrate academic research evolution in interdisciplinary fields: the case of Italian design

PONE-D-24-17407R1

Dear Dr. Barla,

We’re pleased to inform you that your manuscript has been judged scientifically suitable for publication and will be formally accepted for publication once it meets all outstanding technical requirements.

Kind regards,

Diego R. Amancio

Academic Editor

PLOS ONE

Additional Editor Comments (optional):

Reviewers' comments:

Reviewer's Responses to Questions

**Comments to the Author**

1. If the authors have adequately addressed your comments raised in a previous round of review and you feel that this manuscript is now acceptable for publication, you may indicate that here to bypass the “Comments to the Author” section, enter your conflict of interest statement in the “Confidential to Editor” section, and submit your "Accept" recommendation.

Reviewer #1: All comments have been addressed

Reviewer #2: All comments have been addressed

2. Is the manuscript technically sound, and do the data support the conclusions?

Reviewer #1: Yes

Reviewer #2: Yes

3. Has the statistical analysis been performed appropriately and rigorously? 

Reviewer #1: N/A

Reviewer #2: N/A

4. Have the authors made all data underlying the findings in their manuscript fully available?

Reviewer #1: Yes

Reviewer #2: Yes

5. Is the manuscript presented in an intelligible fashion and written in standard English?

Reviewer #1: Yes

Reviewer #2: Yes

6. Review Comments to the Author

Reviewer #1: The authors have addressed the suggestions made in the first round of revision. I therefore have no further suggestions to make. I wish you success with your projects.

Reviewer #2: Dear Authors,

your answers to my comments have been satisfied. i suggest to publish and to continue your research that is interesting for the scientific community

7. PLOS authors have the option to publish the peer review history of their article (what does this mean?). If published, this will include your full peer review and any attached files.

Reviewer #1: **Yes: **Marcio Pereira Basilio

Reviewer #2: **Yes: **Elio Borgonovi

---

## [Editor Report · Acceptance letter]

3 Dec 2024

PONE-D-24-17407R1 

PLOS ONE

Dear Dr. Barla, 

I'm pleased to inform you that your manuscript has been deemed suitable for publication in PLOS ONE. Congratulations! Your manuscript is now being handed over to our production team.

Kind regards, 

on behalf of

Dr. Diego R. Amancio 

Academic Editor

PLOS ONE